# Enhanced Photovoltaic Properties of Perovskite Solar Cells by Employing Bathocuproine/Hydrophobic Polymer Films as Hole-Blocking/Electron-Transporting Interfacial Layers

**DOI:** 10.3390/polym13010042

**Published:** 2020-12-24

**Authors:** Guan-Zhi Liu, Chi-Shiuan Du, Jeng-Yue Wu, Bo-Tau Liu, Tzong-Ming Wu, Chih-Feng Huang, Rong-Ho Lee

**Affiliations:** 1Department of Chemical Engineering, National Chung Hsing University, Taichung 402, Taiwan; as798320@gmail.com (G.-Z.L.); justduit0831@gmail.com (C.-S.D.); s0916871303@gmail.com (J.-Y.W.); HuangCF@dragon.nchu.edu.tw (C.-F.H.); 2Department of Chemical and Materials Engineering, National Yunlin University of Science and Technology, Yunlin 64002, Taiwan; 3Department of Materials Science and Engineering, National Chung Hsing University, Taichung 402, Taiwan; tmwu@nchu.edu.tw

**Keywords:** bathocuproine, methylammonium lead iodide, electron-transporting interfacial layer, perovskite solar cells

## Abstract

In this study, we improved the photovoltaic (PV) properties and storage stabilities of inverted perovskite solar cells (PVSCs) based on methylammonium lead iodide (MAPbI_3_) by employing bathocuproine (BCP)/poly(methyl methacrylate) (PMMA) and BCP/polyvinylpyrrolidone (PVP) as hole-blocking and electron-transporting interfacial layers. The architecture of the PVSCs was indium tin oxide/poly(3,4-ethylenedioxythiophene):polystyrenesulfonate/MAPbI_3_/[6,6]-phenyl-C_61_-butyric acid methyl ester/BCP based interfacial layer/Ag. The presence of PMMA and PVP affected the morphological stability of the BCP and MAPbI_3_ layers. The storage-stability of the BCP/PMMA-based PVSCs was enhanced significantly relative to that of the corresponding unmodified BCP-based PVSC. Moreover, the PV performance of the BCP/PVP-based PVSCs was enhanced when compared with that of the unmodified BCP-based PVSC. Thus, incorporating hydrophobic polymers into BCP-based hole-blocking/electron-transporting interfacial layers can improve the PV performance and storage stability of PVSCs.

## 1. Introduction

Organometal halide perovskites, particularly methylammonium lead iodide (CH_3_NH_3_PbI_3_, MAPbI_3_) and methylammonium lead bromide (CH_3_NH_3_PbBr_3_, MAPbBr_3_), are highly suitable for use in solar cells because of their excellent photovoltaic (PV) properties (large absorption coefficients in the Vis to near-infrared (NIR) region, long diffusion lengths of charge carriers, excellent charge mobility, and high photoconversion efficiencies (PCEs)) [1,2,3,4,5,6]. Perovskite solar cells (PVSCs), having planar or mesoscopic structures, can be fabricated through solution processing with thermal treatment at low temperatures [6,7,8]. Many efforts have been made to improve the PV properties of PVSCs; for example, by designing new perovskite materials with high photo-electron conversion [9,10], interfacial engineering between the perovskite layer and the electron- or hole-transporting layer (ETL or HTL) [11,12,13,14], controlling the crystal growth of the perovskite [15,16,17,18,19,20], and adding suitable amounts of halide, cationic, organic, and polymer additives in the perovskite layer [21,22,23,24,25,26,27,28,29].

In addition to a high PCE, a PVSC must possess high operational stability if it is to find wide applicability [26,30]. Incorporating an inorganic material [27,28] or hydrophobic polymer under the cathode layer can be an effective method to protect the PVSC from the permeation of oxygen and humidity [29]. For example, hydrophobic poly(methyl methacrylate) (PMMA) has been incorporated into PVSCs to enhance their operational and storage stabilities [30,31,32,33]. Kundu et al. reported a PVSC featuring an HTL comprising of poly(3-hexylthiophene) (P3HT) and PMMA; the PMMA matrix imparted good resistance to the permeation of moisture, resulting in a large enhancement of the stability of the cell [30]. Habisreutinger et al. found that depositing PMMA on top of a P3HT/single-walled carbon nanotube (SWNT) nanohybrid–based HTL filled the voids within the P3HT/SWNT nanohybrid and blocked the contact of the Ag-based cathode with the MAPbI_3_ layer; incorporating the PMMA layer enhanced the shunt resistance, open-circuit voltage (*V*_OC_), and fill factor (FF) of the PVSC, while also inhibiting the permeation of moisture into the MAPbI_3_ layer and increasing the storage stability [32]. Furthermore, adding PMMA into two-dimensional layered MAPbI_3_ intermediates has led to self-assembly into three-dimensional perovskite crystal grains featuring a coating of PMMA at the crystal grain boundaries; this bilayer structure inhibited the permeation of moisture and enhanced the stability of MAPbI_3_ [33]. Moreover, the presence of a PMMA film can decrease the trap density by compensating electronically for iodide vacancies along the boundary; this approach can minimize charge recombination and improve the values of *V*_OC_ of PVSCs [33].

In addition to PMMA, the hydrophobic polymer polyvinylpyrrolidone (PVP) has also been used as an interfacial material between the ETL and the cathode to improve PV performance. The quality of the ETL–cathode interface can have a dramatic effect on the electron transport and extraction in the PVSC. Incorporating PVP has promoted electron transport across the perovskite–cathode interface when using a [6,6]-phenyl-C_61_-butyric acid methyl ester (PC_61_BM)–based ETL. Moreover, the presence of a dipole layer after the addition of PVP can enhance the built-in potential across the cell, thereby favoring charge transport from the ETL to the cathode and improving the PV properties [34]. In addition, hydrophobic PVP has been used as an additive in the perovskite layer to improve PV performance by protecting the perovskite crystals from the damaging effects of moisture; the acylamino groups of PVP enhanced the electron density at the perovskite surface and, thereby, decreased the surface energy and stabilized the perovskite layer; the resulting PVSC displayed a high PCE and excellent moisture-stability [35]. Furthermore, strong interactions between Pb(II) ions and the C=O groups in PVP can lead to the nuclei distributing uniformly along the PVP chains, resulting in compact MAPbI_3_ films; consequently, PVP-added perovskite layers can possess crystalline structures that enhance the PCE-stability of their PVSCs [36,37,38,39].

In inverted PVSCs, the fullerene derivative PC_61_BM is usually employed as an electron extraction layer and ETL at the perovskite–cathode interface. Introducing a bathocuproine (BCP)-based hole-blocking/electron-transporting interfacial layer between the PC_61_BM layer and the cathode can enhance a PVSC’s FF and short-circuit current density (*J*_SC_) by filling voids and improving the surface morphology of the PC_61_BM layer [6,40]. The polar functional groups of BCP can enhance the value of *V*_OC_ of a PVSC by effectively enhancing its built-in potential [41]. Moreover, the presence of BCP prevents Ag atoms from diffusing from the cathode to the MAPbI_3_ layer, thereby enhancing the operational stability of the PVSCs [42]. In this present study, we measured the PV properties and storage stabilities of MAPbI_3_-based inverted PVSCs incorporating BCP/PMMA and BCP/PVP composites as their hole-blocking/electron-transporting interfacial layers. The architecture of our PVSCs (Figure 1) was indium tin oxide (ITO)/poly(3,4-ethylenedioxythiophene):polystyrenesulfonate (PEDOT:PSS)/MAPbI_3_/PC_61_BM/BCP based interfacial layer/Ag. Incorporating PMMA into the BCP layer enhanced the storage stability of our PVSC, while PVP in the BCP layer facilitated electron transport at the perovskite–cathode interface. We used scanning electron microscopy (SEM) and X-ray diffractometry (XRD) to examine the morphologies and crystal structures of the resulting perovskite layers, and atomic force microscopy (AFM) to observe the morphologies of the BCP/PMMA and BCP/PVP layers. The hydrophobicity/hydrophilicity of the BCP/PMMA and BCP/PVP layers were determined using a contact angle (CA) meter. We found that the BCP/PMMA and BCP/PVP interfacial layers enhanced the PV properties, the PCEs, and the storage stabilities of their MAPbI_3_-based PVSCs.

## 2. Experimental Details

### 2.1. Materials and Instrumentation

Methylamine (CH_3_NH_2_), lead iodide (PbI_2_), BCP, PMMA (*M*_w_ = 15,000), and PVP (*M*_w_ = 10,000) were purchased from Sigma–Aldrich (St. Louis, MO, USA), Acros, (Fukuoka, Japan) and TCI Chemical (Tokyo, Japan), and used without purification. PC_61_BM was purchased from NANO-C (Beijing, China). Isopropanol, γ-butyrolactone (GBL), dimethylsulfoxide (DMSO), toluene, and *o*-dichlorobenzene (*o*-DCB) were distilled over appropriate drying agents prior to use.

Fourier transform infrared (FTIR) spectra were recorded using a HORIBA FT-720 FTIR spectrometer (HORIBA Inc., Tainan City, Taiwan). Differential scanning calorimetry (DSC-2010, TA Instruments, New Castle, DE, USA) was used to determine the glass transition temperatures (*T*_g_) of the BCP/PMMA and BCP/PVP blend films under a N_2_ atmosphere (scanning rate: 10 °C min^−1^). UV–Vis absorption spectra were recorded using a Hitachi U3010 UV–Vis spectrometer (Hitachi High-Tech Co., Tokyo, Japan). Photoluminescence (PL) spectra were recorded using a Hitachi F-4500 fluorescence spectrophotometer. AFM images of BCP/PMMA and BCP/PVP blend films coating the surface of the ETL (PC_61_BM) were recorded using a Seiko SII SPA400 (Chiba, Japan) atomic force microscope operated in tapping mode. Cold field emission scanning electron microscopy (FESEM) images of the MAPbI_3_ layer were recorded using a Hitachi S-4800 microscope (Integrated Service Tech. Inc., Hinchu, Taiwan). Powder XRD of the perovskite layer was measured using a Shimadzu SD-D1 instrument (Shimadzu Scientific Instrument Co., Taipei, Taiwan) and a Cu target. The CAs of water droplets on the BCP/PMMA and BCP/PVP films were determined using a Kyowa Drop Master optical CA meter (Applied Trentech Inc., Taipei, Taiwan).

### 2.2. Fabrication and Characterization of PVSCs

The architecture of the PVSCs was ITO-coated glass/PEDOT:PSS/MAPbI_3_/PC_61_BM/BCP:PMMA or BCP:PVP/Ag (100 nm). The photoactive area of each device was 0.24 cm^2^. ITO-coated glass, having a sheet resistance of 20 Ω square^−1^, was obtained from Luminescence Tech (New Taipei City, Taiwan); PC_61_BM was procured from Nanocarbon (Beijing, China). The glass substrates featuring patterned ITO electrodes were washed well and then cleaned using O_2_ plasma. PEDOT:PSS (AI4083, Heraeus Clevios Co., Hanau, Germany) was spin-coated onto the ITO layer. The sample was heated at 110 °C for 30 min. Methylammonium iodide (MAI; 0.200 g, 1.25 mmol) and PbI_2_ (0.580 g, 1.25 mmol) were stirred in a mixture of GBL and DMSO (1:1, *v*/*v*; 1 mL) to obtain a MAPbI_3_ solution. The MAPbI_3_ solution (0.2 mL) was deposited on the surface of the PEDOT:PSS layer through two consecutive spin-coating processes at 1000 and 3000 rpm for 10 and 30 s, respectively. During the second spin-coating process, toluene (0.5 mL) was drop-cast onto the substrate, which was then dried on a hot plate (80 °C, 5 min). A solution (0.2 mL) of PC_61_BM (20 mg mL^−1^) in *o*-DCB was coated on top of the MAPbI_3_ layer; a solution (0.3 mL) of BCP or a BCP/polymer (PMMA or PVP) blend in isopropanol (0.5 mg mL^−1^) was then coated on the PC_61_BM layer. The Ag cathode was thermally deposited onto the BCP/PMMA or BCP/PVP layer under high vacuum. The PV parameters of the PVSCs were determined using a programmable electrometer (Keithley 2400, Keithley Instruments, Inc., Cleveland, OH, USA) under illumination with AM1.5 light from a solar simulator (NewPort Oriel 96000, Newport Corporation, Taipei, Taiwan) at an intensity of 100 mW cm^−2^.

## 3. Results and Discussion

### 3.1. Chemical Structures and Thermal Properties of BCP/PMMA and BCP/PVP Blends

The functional groups in the BCP/PMMA and BCP/PVP blends were characterized using FTIR spectroscopy. FTIR spectra of BCP, PMMA, PVP, BCP/PMMA (5:1, *w*/*w*), and BCP/PVP (5:1, *w*/*w*) are provided in Figure 2. In the spectra of BCP, PMMA, and PVP, the signals for C–H stretching appeared in the range 2300–3200 cm^−1^ and those for C–H bending in the CH_3_ and CH_2_ units at 1450–1600 cm^−1^. In the spectrum of BCP, the signal for =C–H bending of the aromatic ring appeared at 707 cm^−1^; the signal of the imino (C=N) group appeared at 1716 cm^−1^; and the signals for C=C stretching in the aromatic ring appeared at 1488 and 1569 cm^−1^. In the spectra of PMMA and PVP, the signals for C=O stretching appeared at 1733 and 1656 cm^−1^, respectively; for PMMA, the signals for the C–O moieties of the ester groups appeared at 1147 and 1193 cm^−1^; for PVP, the signal for C–N stretching appeared at 1290 cm^−1^. Furthermore, the signals of BCP, PMMA, and PVP appeared in the spectra of the BCP/PMMA and BCP/PVP blends.

Figure 3 displays DSC thermograms of BCP, BCP/PMMA (5:1, *w*/*w*), and BCP/PVP (5:1, *w*/*w*). The values of *T*_m_ for BCP, BCP/PMMA, and BCP/PVP samples were 289, 279, and 283 °C, respectively. The lower value of *T*_m_ of the BCP/PMMA blend implies that its miscibility was greater than that of the BCP/PVP blend.

### 3.2. Optical Properties of MAPbI_3_ Perovskite Film

We recorded UV–Vis absorption and photoluminescence (PL) spectra to examine the optical properties of the MAPbI_3_ layer (Appendix A). The absorption onset of the MAPbI_3_ layer (ca. 780 nm) suggested an optical band gap of 1.6 eV [43]. The maximal PL wavelength of the MAPbI_3_ layer appeared near 768 nm, when excited at a wavelength of 510 nm [44].

### 3.3. SEM Images of MAPbI_3_ Layers Coated with Various Electron-Transporting Interfacial Films

Figure 4 presents SEM images of the MAPbI_3_ perovskite film. In addition to the presence of grain boundaries between crystals, the crystal grains were distributed uniformly in the MAPbI_3_ perovskite film, without other defects. The largest crystal grain had a size of approximately 210 nm. Such a high-quality MAPbI_3_ layer is generally favorable for PVSCs displaying high PV performance. Figure 5 displays cross-sectional SEM images of ITO/PEDOT:PSS/MAPbI_3_/PC_61_BM structures coated with electron-transporting interfacial layers of BCP, BCP/PMMA, and BCP/PVP films. The thicknesses of the MAPbI_3_ layers ranged from 275.6 to 290.6 nm. We suspected that the dense packing of the grain crystals of the MAPbI_3_ layer would minimize grain boundary defects and enhance the charge transfer capacity. These MAPbI_3_-based photoenergy conversion layers possessed good film quality. In addition, the thicknesses of the PC_61_BM/BCP/polymer (PMMA or PVP) structures ranged from 275.6 to 290.6 nm. These SEM images revealed that the BCP/polymer composites filled the voids in the PC_61_BM layer during the spin-coating process. Therefore, only a single layer containing PC_61_BM and the BCP/polymer composite appeared on the surface of the MAPbI_3_ layer. The good passivation of the BCP/polymer composite on the PC_61_BM layer would potentially facilitate electron transport from PC_61_BM to the cathode.

### 3.4. XRD Images of MAPbI_3_ Perovskite Films Coated with Various Hole-Blocking/Electron-Transporting Interfacial Films

To investigate the effect of the hole-blocking/electron-transporting interfacial layers on the storage-stability of the MAPbI_3_ film, we used XRD to study the crystal structures of the MAPbI_3_ films after storage at 30 °C and 35% relative humidity for 10 days. Figure 6 presents the XRD patterns of the ITO/PEDOT:PSS/MAPbI_3_/PC_61_BM structures coated with electron-transporting interfacial layers of BCP, BCP/PMMA (5:1, *w*/*w*), and BCP/PVP (5:1, *w*/*w*). The patterns of the MAPbI_3_ films featured diffraction peaks at 14.2, 28.4, and 31.6°, corresponding to the (110), (220), and (310) phases, respectively [45,46], suggesting tetragonal perovskite structures having lattice constants *a* and *b* of 8.883 Å and *c* of 12.677 Å [45]. Figure 6a reveals that MAPbI_3_ dissociated partially into MAI and PbI_2_ after storage under the ambient conditions for 10 days, with a diffraction peak for PbI_2_ appearing at 13° [45]. For the MAPbI_3_ film coated with the BCP/PMMA blend film, the intensity of this diffraction peak at 13° was suppressed significantly relative to that of the MAPbI_3_ coated with BCP (Figure 6b). Thus, the addition of PMMA in the BCP layer inhibited the permeation of moisture into the MAPbI_3_ layer to prevent its dissociation. In contrast, when compared with the MAPbI_3_ film coated with the layer of BCP, the intensity of the PbI_2_ diffraction peak at 13° was higher for the MAPbI_3_ film coated with the BCP/PVP blend film after storage under ambient conditions for 5 and 10 days (Figure 6c). Thus, the MAPbI_3_ film coated with the BCP/PVP blend film was not resistant toward the permeation of moisture, suggesting poor compatibility between BCP and PVP. Moreover, the crystal sizes of the MAPbI_3_ films coated with BCP, BCP/PMMA, and BCP/PVP layers after storage at 30 °C and 35% relative humidity for 0, 5, and 10 days are summarized in Appendix A. The crystal sizes of the MAPbI_3_ films decreased as the storage time increased, which corresponded to the degradation of the perovskite crystals. Moisture entering MAPbI_3_ caused the decreased crystals sizes.

### 3.5. AFM Images and CAs of BCP/PMMA and BCP/PVP Composite Films

We used AFM to study the surface morphologies of the BCP/PMMA and BCP/PVP hole-blocking/electron-transporting interfacial layers. Appendix A display topographic and phase images of films of BCP and the BCP/PMMA and BCP/PVP composites, recorded after thermal treatment at 80 °C for 5 min. Appendix A summarizes the surface roughness of these BCP, BCP/PMMA, and BCP/PVP films. The topographic images in Appendix A reveal that the surface roughness of the BCP films was not enhanced after the addition of PMMA. No phase separation was evident in the phase images of the composite films. Thus, BCP and PMMA exhibited good compatibility. In contrast, significant phase separation was evident in the phase images of the BCP/PVP composite films (Appendix A), suggesting poor compatibility between BCP and PVP. Nevertheless, the surface roughness of the BCP/PVP films were not higher than those of the BCP/PMMA films.

We also used AFM to determine the stability of the surface morphologies of the films of BCP and the BCP/PMMA and BCP/PVP composites. Figure 7, Figure 8 and Figure 9 display AFM images of the BCP, BCP/PMMA (5:1, *w*/*w*), and BCP/PVP (5:1, *w*/*w*) films, respectively, after storage at 30 °C and 35% relative humidity for 0, 5, and 10 days. The morphology of BCP film was modified only slightly after storage for 5 days, but it changed significantly after 10 days (Figure 7). The morphology of the BCP/PMMA film (Figure 8) had changed only slightly after storage for 10 days, consistent with the good compatibility of BCP and PMMA. In other words, the BCP/PMMA film had high morphological stability. In contrast, the stability of the morphology of the BCP/PVP film was much poorer than those of the BCP film (Figure 9). Some of the BCP had separated from the BCP/PVP film, leading to the formation of BCP-based crystals after storage for 5 days, with a higher density appearing after storage for 10 days. Unlike the BCP and BCP/PMMA films, the surface roughness of the BCP/PVP film was enhanced significantly upon increasing the storage time under ambient conditions.

To examine the mechanism behind the morphological changes of the composite films under ambient conditions, we measured the hydrophobicity/hydrophilicity of the BCP/PMMA and BCP/PVP films using a CA meter. Appendix A display photographs of water droplets on the BCP, BCP/PMMA, and BCP/PVP films that had been stored at 30 °C and 35% relative humidity for 0, 5, and 10 days. Appendix A lists the CAs of the BCP, BCP/PMMA, and BCP/PVP films. Initially, the CA of the BCP film was 74.5°, suggesting low hydrophobicity; the CA decreased upon increasing the storage time, consistent with an increase in the permeation of moisture into this film and the significant change in the morphology of BCP film observed in Figure 7. The CA of the BCP film increased upon increasing the content of PMMA. Moreover, the CA of the BCP/PMMA (5:1, *w*/*w*) film did not decrease with increasing the storage time. The presence of the highly hydrophobic PMMA prevented the intrusion of moisture into the BCP film, consistent with the high morphological stability observed for the BCP/PMMA composite film in Figure 8. In contrast, the CA decreased upon increasing the PVP content in the BCP/PVP composite films. Furthermore, the CA of the BCP/PVP (5:1, *w*/*w*) film decreased significantly upon increasing the storage time. The presence of PVP, with low hydrophobicity, enhanced the permeation of moisture into the BCP film. As a result, phase separation of BCP and PVP occurred, promoting the formation of BCP-based crystals in the BCP/PVP film, as observed in Figure 9.

## 4. PV Characteristics

Figure 10 presents the best PV performance of PVSCs featuring the BCP, BCP/PMMA, and BCP/PVP films as hole-blocking/electron-transporting interfacial layers; the statistical values of *V*_OC_, *J*_SC_, FF, and PCE are summarized in Table 1. Four runs of PV evaluation tests were performed for each PVSC sample. For PVSC I, prepared with BCP as the electron-transporting interfacial layer, a value of *V*_OC_ of 0.90 V, a value of *J*_SC_ of 21.2 mA cm^−2^, an FF of 0.62, and a PCE of 11.78% were obtained. Compared with the BCP-based PVSC, the values of *V*_OC_ and the FFs of the PVSCs incorporating the BCP/PMMA blend films (PVSCs II–IV) did not change significantly upon varying the PMMA content; the values of *J*_SC_ and the PCEs did, however, decrease slightly upon increasing the PMMA content, consistent with the lower BCP content in the electron-transporting interfacial layer. Decreasing the BCP content inhibited electron transport from the MAPbI_3_ layer to the cathode. The lowest values of *J*_SC_ and PCE were those of PVSC IV incorporating 40 wt.% PMMA. In contrast, the values of *V*_OC_, *J*_SC_, and PCE of the PVSCs incorporating BCP/PVP blend films (PVSCs V–VII) increased upon increasing the PVP content, and were greater than those of the BCP-based PVSC. The highest values of *V*_OC_ (0.92 V), *J*_SC_ (21.72 mA cm^−2^), FF (0.62), and PCE (12.41%) were obtained for PVSC VI, prepared with BCP/PVP (5:1, *w*/*w*) as the hole-blocking/electron-transporting interfacial layer. The presence of PVP at the interface between PC_61_BM and the cathode might have induced the formation of a dipole layer, thereby enhancing the built-in potential across the cell and facilitating electron transport from the perovskite layer to the cathode [45]. As a result, the PV properties of the BCP/PVP-based PVSCs were enhanced. Nevertheless, a larger PVP content in the BCP/PVP blend film did not improve the PV properties of PVSC VII, which provided a PCE lower than that of PVSC I. The poor compatibility of BCP and PVP would inhibit electron transport from PC_61_BM to the cathode. Consequently, the PV performance of the BCP/PVP (5:2, *w*/*w*)–based PVSC VII was poor. Appendix A reveals that the PV properties of our PVSCs reported herein are comparable with those of similarly structured PVSCs reported previously in the literature.

Figure 11 displays the external quantum efficiency (EQE) spectra of the PVSCs incorporating films of BCP, BCP/PMMA, and BCP/PVP. Similar EQE profiles of MAPbI_3_-based PVSCs have been reported several times previously [11,47,48]. The partial photo-response near 400 nm arose from the absorption of PC_61_BM [49]. Relative to the BCP-based PVSC I, the EQEs decreased upon increasing the amount of PMMA in the BCP/PMMA-based PVSC-II, PVSC-III, and PVSC-IV. A greater PMMA content in the BCP/PMMA blend film inhibited the transport of electrons from the MAPbI_3_ layer to the cathode. As a result, the EQEs decreased when the PVSCs had higher PMMA contents. In contrast, the EQEs of the BCP/PVP-based PVSC V, PVSC-VI, and PVSC-VII were greater than those of the BCP-based PVSC I. The presence of PVP at the interface between the PC_61_BM layer and the cathode might have promoted the transport of electrons from the PC_61_BM layer to the cathode. Nevertheless, a larger PVP content in the BCP/PVP blend film did not enhance the EQE of PVSC VII—it was lower than that of PVSC I. Moreover, the *J*_SC_ values of the PVSCs calculated from EQE spectra are summarized in Table 1. The *J*_SC_ values are in relatively good agreement with the values measured under irradiation of AM 1.5 G light at 100 mW cm^−2^.

The storage stability of the BCP-, BCP/PMMA-, and BCP/PVP-based PVSCs measured at 30 °C and 35% relative humidity is displayed in Figure 12. The PCE-stabilities of the BCP/PMMA-based PVSC II and PVSC III were superior to that of the BCP-based PVSC-I. Higher PCE-stabilities were observed for the BCP/PMMA-based PVSCs upon increasing their PMMA content, consistent with the higher hydrophobicity of PMMA preventing moisture from entering the BCP film. Nevertheless, the PCE-stability of PVSC IV was lower than those of PVSC II and PVSC III, presumably because of the poorer PV properties of PVSC IV with its higher PMMA content. In contrast, the PCE-stabilities of the BCP/PVP-based PVSCs were poorer than that of the BCP-based PVSC-I, with lower PCE-stability observed upon increasing the PVP content. The phase separation of BCP and PVP, and the high hydrophilicity of PVP, promoted the intrusion of moisture into the BCP film. Consequently, the PVSCs incorporating the BCP/PVP composite film exhibited poor PCE-stability.

In addition, the time dependence of the *V*_OC_ and *J*_SC_ of the PVSC I, PVSC III, and PVSC VI (measured at 30 °C and 35% relative humidity) is shown in Appendix A. The results indicated that the *V*_OC_ values of the PVSCs slightly decreased with increasing storage time, whereas the *J*_SC_ values of the PVSCs decreased considerably with increasing storage time. The intrusion of moisture into the MAPbI_3_ layer enhanced the impendence of the cells. As a result, the *J*_SC_ values of the PVSCs decreased. The *J*_SC_ dominated the PCE degradation of PVSCs.

To study the influence of moisture permeation on the stability of the PVSC device, the storage stability of the BCP-, BCP/PMMA-, and BCP/PVP-based PVSCs was measured at 30 °C and 60% relative humidity; details concerning these measurements are displayed in Appendix A. The stability of the PVSCs at 60% relative humidity was substantially reduced compared with that of the PVSCs at 35% relative humidity, suggesting that the intrusion rate of moisture into the MAPbI_3_ layer is enhanced by increasing the relative humidity. Therefore, the degradation rate of the MAPbI_3_-based PVSCs increased as the cells were stored under higher relative humidity.

## 5. Conclusions

The PV properties and stabilities of MAPbI_3_-based PVSCs incorporating BCP/PMMA and BCP/PVP films as hole-blocking/electron-transporting interfacial layers have been studied. The storage-stability of the BCP/PMMA-based PVSCs was enhanced significantly relative to that of the BCP-based PVSC, but the PV performance decreased slightly after the incorporation of PMMA. The improved storage-stability arose from the greater hydrophobicity and moisture-resistance of the resulting BCP/PMMA layer. A higher content of PMMA in the BCP layer inhibited charge transfer from the MAPbI_3_ layer to the cathode, resulting in a lower short-circuit current density and a less efficient PVSC. In contrast, the PV performance of BCP/PVP-based PVSCs was enhanced relative to that of the BCP-based PVSC, but their storage-stability was poor. The presence of PVP promoted the electron-transporting across the BCP-based interfacial layer to cathode and resulted in PVSCs having higher current densities and PCEs. Nevertheless, poor compatibility between PVP and BCP resulted in the BCP/PVP-based PVSCs displaying poor storage-stability.

## Figures and Tables

**Figure 1 polymers-13-00042-f001:**
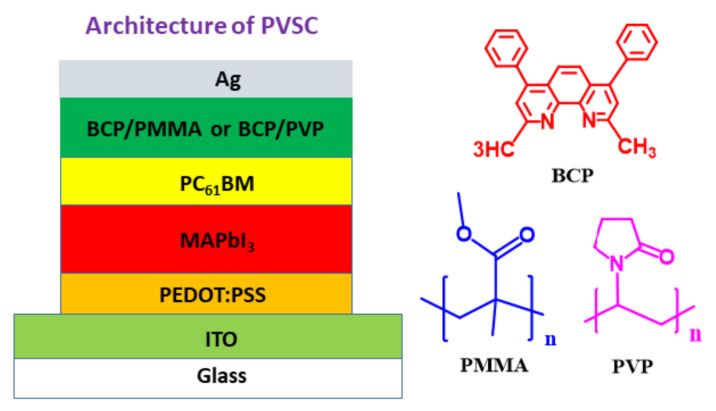
Architecture of the inverted perovskite solar cells (PVSCs).

**Figure 2 polymers-13-00042-f002:**
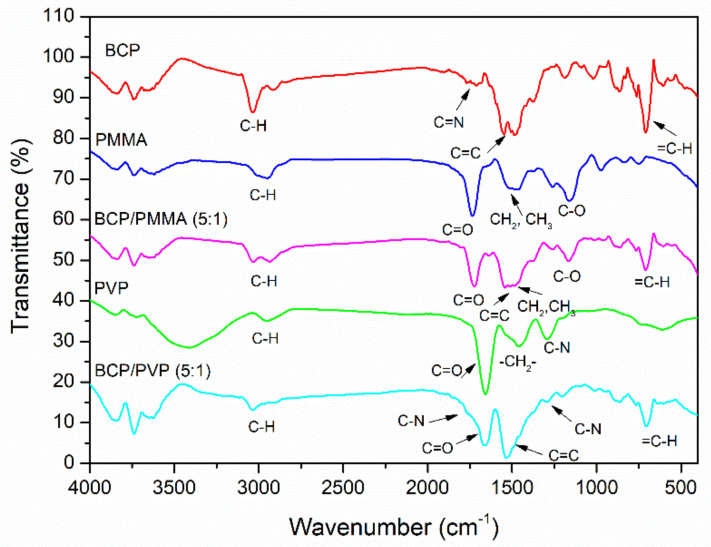
FTIR (Fourier transform infrared) spectra of bathocuproine (BCP), poly(methyl methacrylate) (PMMA), polyvinylpyrrolidone (PVP), BCP/PMMA (5:1, *w*/*w*), and BCP/PVP (5:1, *w*/*w*).

**Figure 3 polymers-13-00042-f003:**
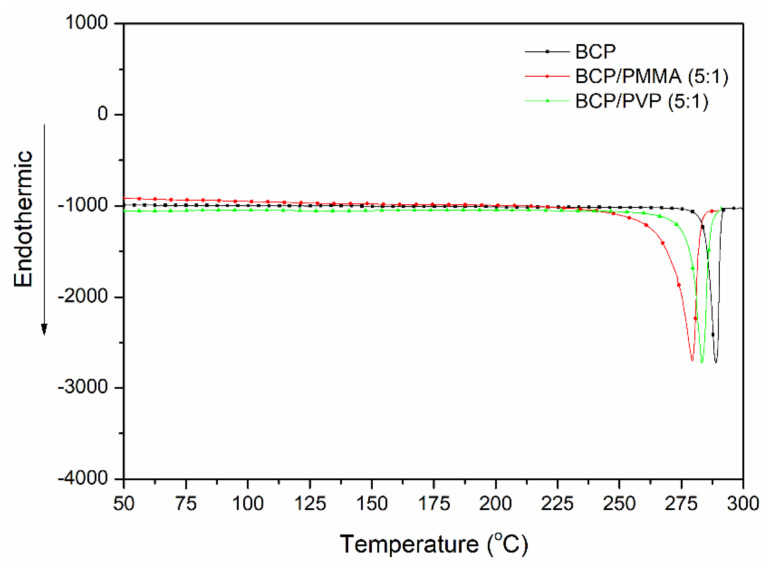
DSC (differential scanning calorimetry) thermograms of BCP, BCP/PMMA (5:1, *w*/*w*), and BCP/PVP (5:1, *w*/*w*).

**Figure 4 polymers-13-00042-f004:**
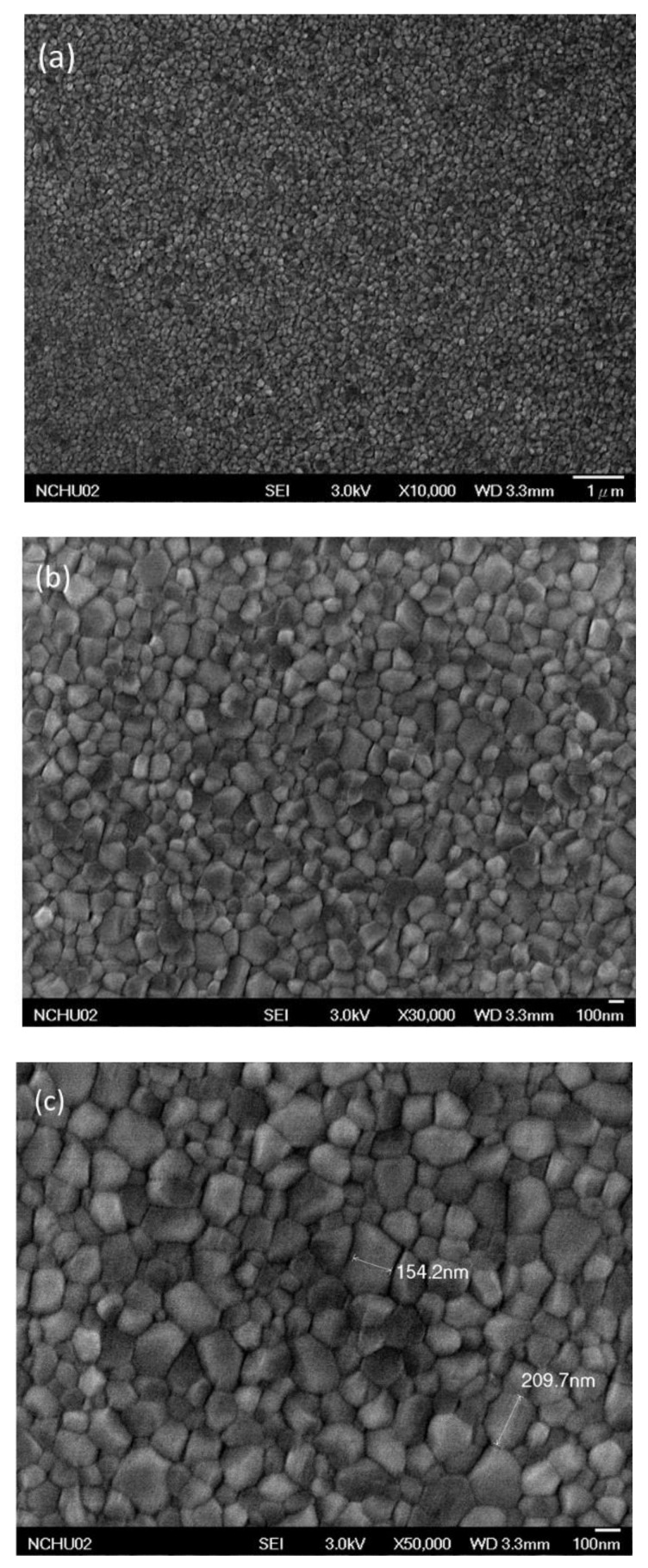
SEM (scanning electron microscopy) images [(**a**) ×10,000; (**b**) ×30,000; (**c**) ×50,000] of a MAPbI_3_ perovskite film coated on the HTL, recorded after thermal annealing (80 °C, 5 min).

**Figure 5 polymers-13-00042-f005:**
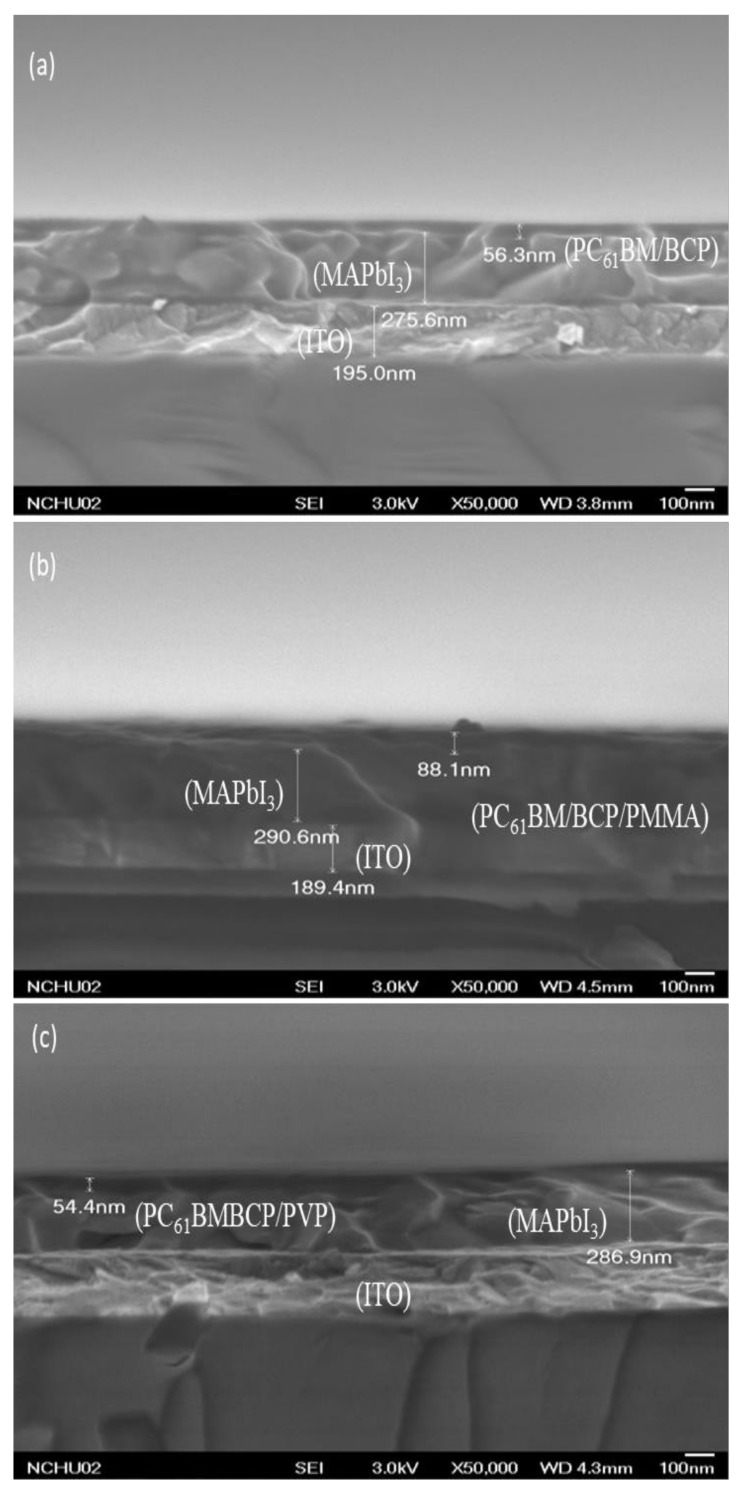
Cross-sectional SEM images of ITO/PEDOT:PSS/MAPbI_3_/PC_61_BM structures coated with hole-blocking/electron-transporting interfacial layers of (**a**) BCP, (**b**) BCP/PMMA (5:1, *w*/*w*), and (**c**) BCP/PVP (5:1, *w*/*w*), recorded after thermal annealing (80 °C, 5 min).

**Figure 6 polymers-13-00042-f006:**
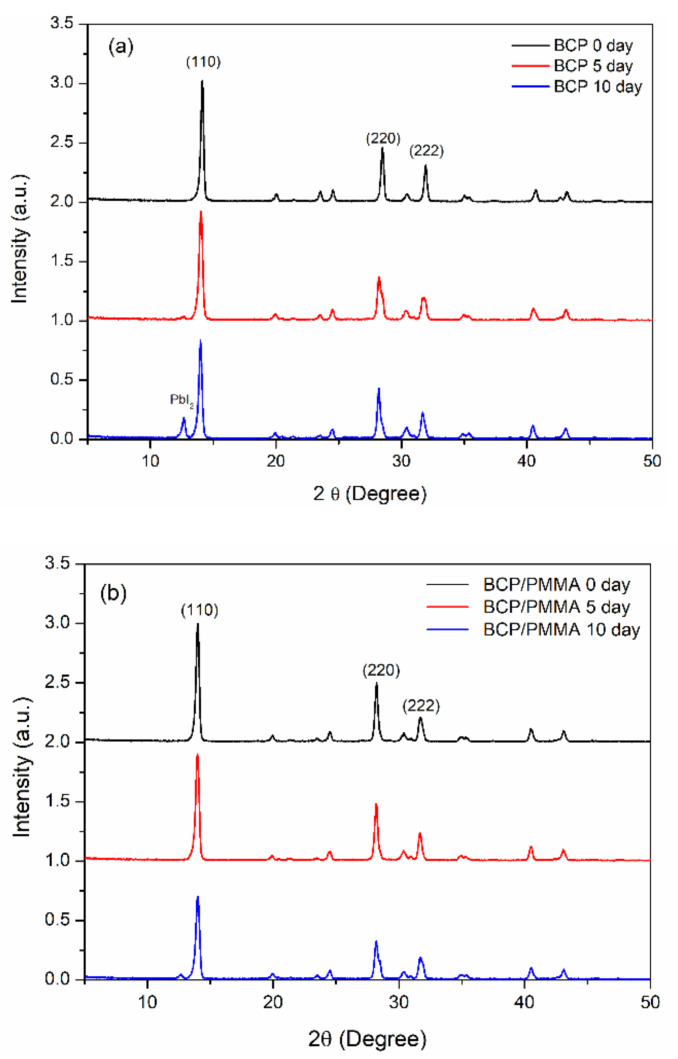
XRD (X-ray diffractometry) patterns of ITO/PEDOT:PSS/MAPbI_3_/PC_61_BM structures coated with hole-blocking/electron-transporting interfacial layers of (**a**) BCP, (**b**) BCP/PMMA (5:1, *w*/*w*), and (**c**) BCP/PVP (5:1, *w*/*w*), recorded after storage at 30 °C and 35% relative humidity for 0, 5, and 10 days.

**Figure 7 polymers-13-00042-f007:**
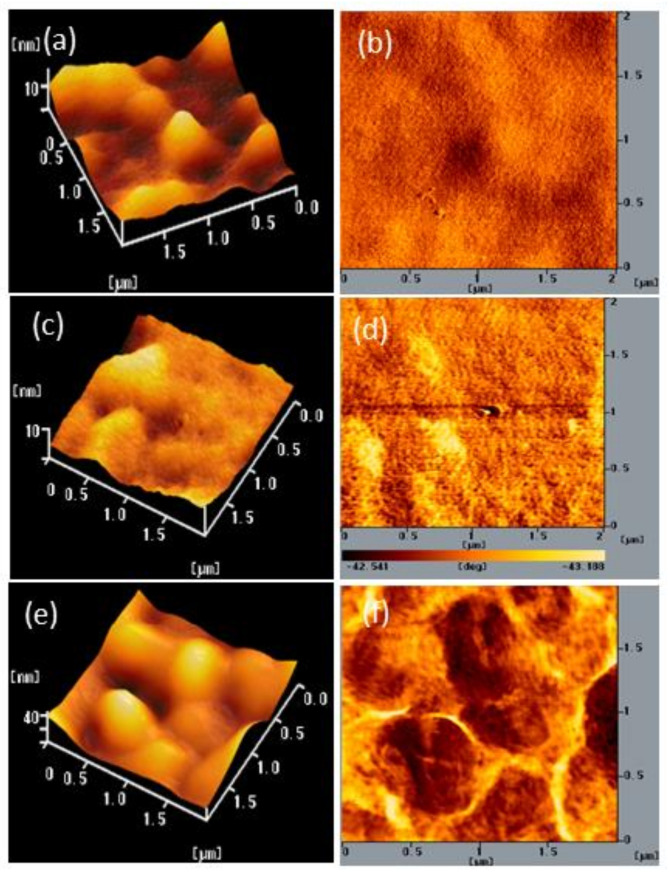
(**a**,**c**,**e**) Topographic and (**b**,**d**,**f**) phase AFM images of a BCP film after storage at 30 °C and 35% relative humidity for (**a**,**b**) 0, (**c**,**d**) 5, and (**e**,**f**) 10 days.

**Figure 8 polymers-13-00042-f008:**
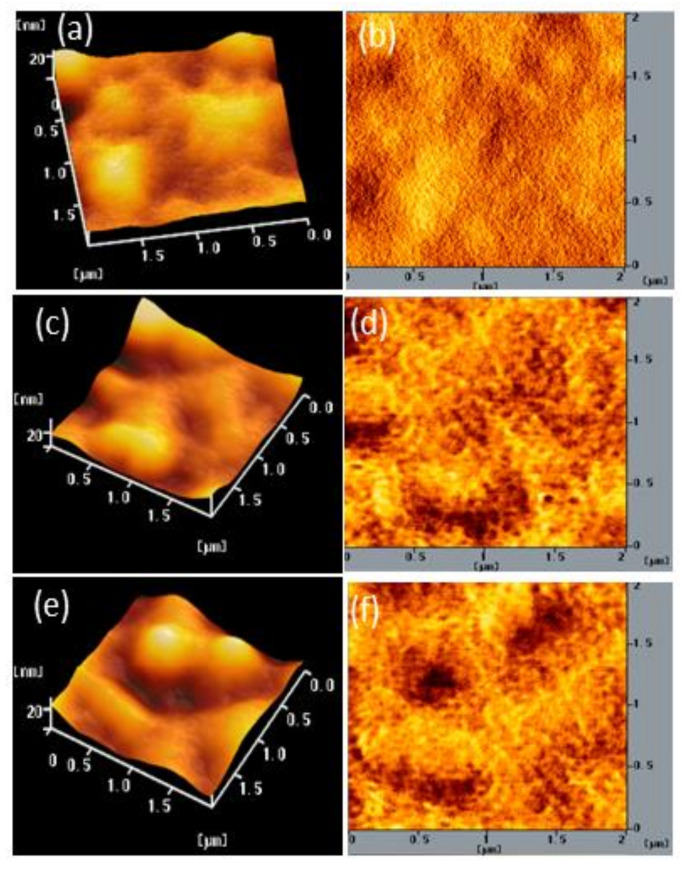
(**a**,**c**,**e**) Topographic and (**b**,**d**,**f**) phase AFM images of a BCP/PMMA (5:1, *w*/*w*) film after storage at 30 °C and 35% relative humidity for (**a**,**b**) 0, (**c**,**d**) 5, and (**e**,**f**) 10 days.

**Figure 9 polymers-13-00042-f009:**
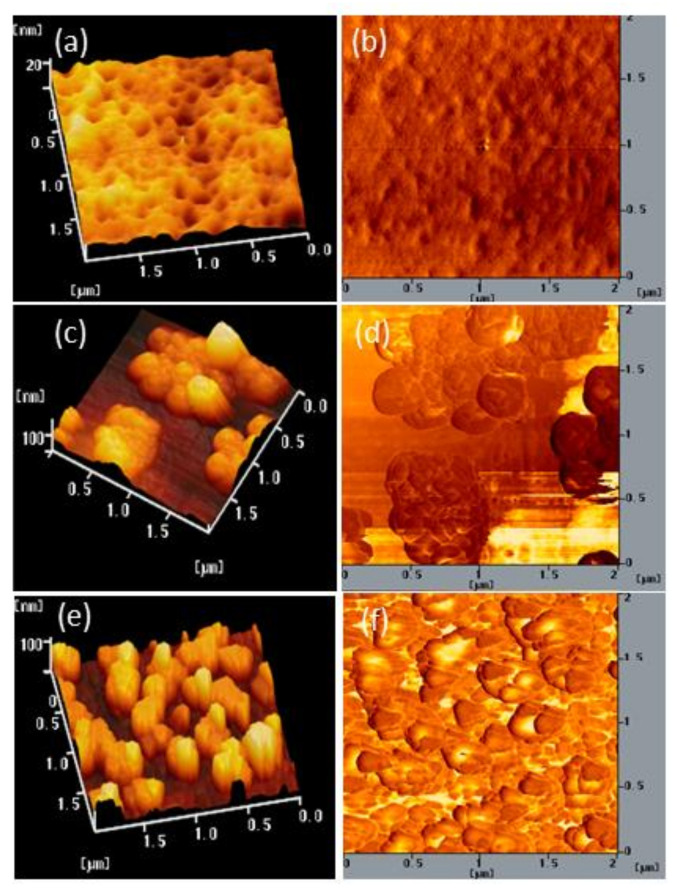
(**a**,**c**,**e**) Topographic and (**b**,**d**,**f**) phase AFM images of a BCP/PVP (5:1, *w/w*) film after storage at 30 °C and 35% relative humidity for (**a**,**b**) 0, (**c**,**d**) 5, and (**e**,**f**) 10 days.

**Figure 10 polymers-13-00042-f010:**
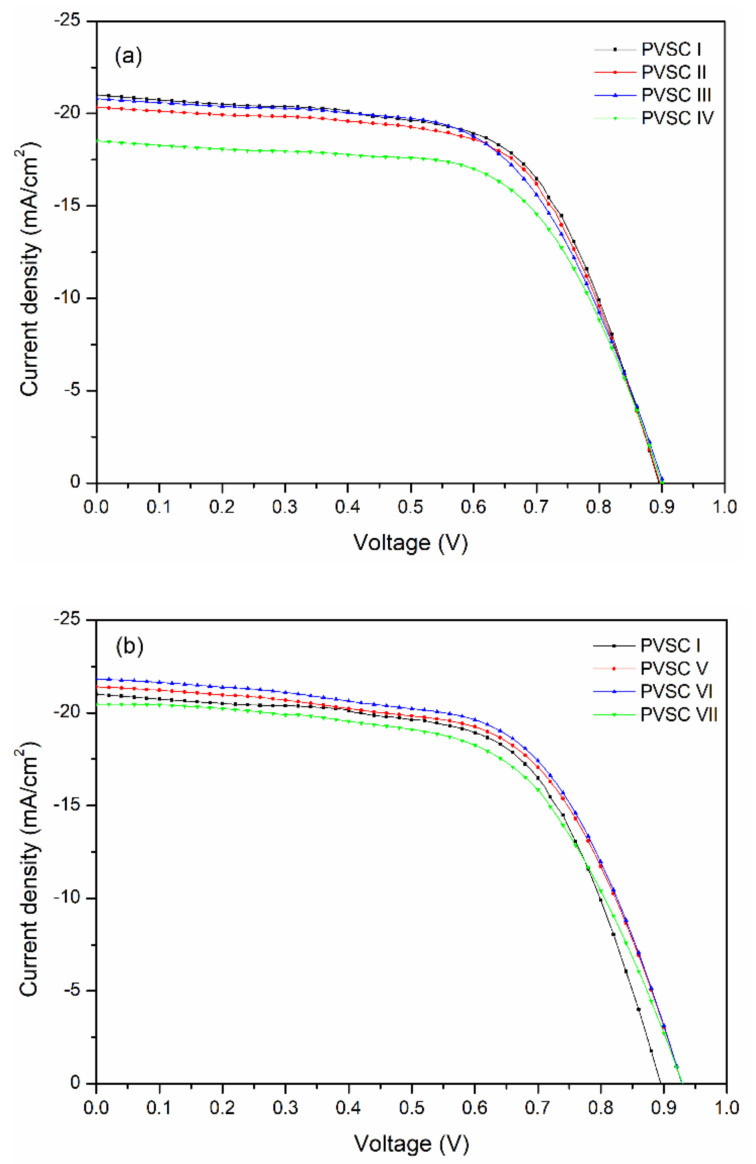
Current density–potential plots of PVSCs incorporating BCP, BCP/PMMA, and BCP/PVP films ((**a**) BCP (PVSC I), BCP/PMMA (PVSC II-IV); (**b**) BCP/PVP (PVSC V-VII)), illuminated under AM 1.5G light at 100 mW cm^−2^.

**Figure 11 polymers-13-00042-f011:**
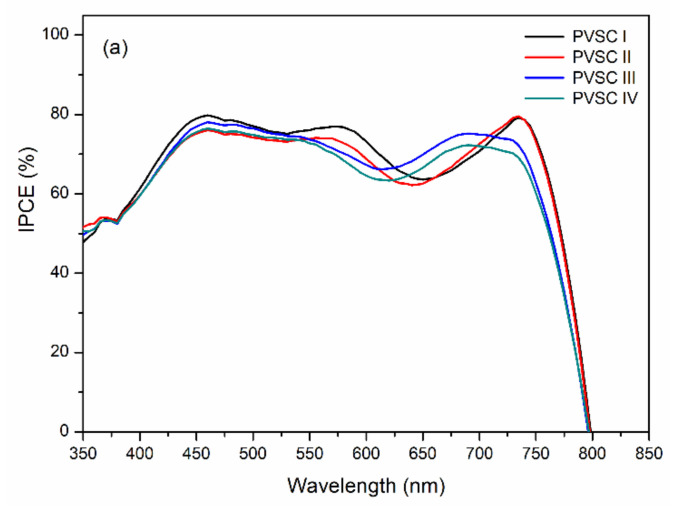
EQE (external quantum efficiency) spectra of PVSCs incorporating films of BCP, BCP/PMMA, and BCP/PVP films ((**a**) BCP (PVSC I), BCP/PMMA (PVSC II-IV); (**b**) BCP/PVP (PVSC V-VII)), recorded under monochromatic irradiation.

**Figure 12 polymers-13-00042-f012:**
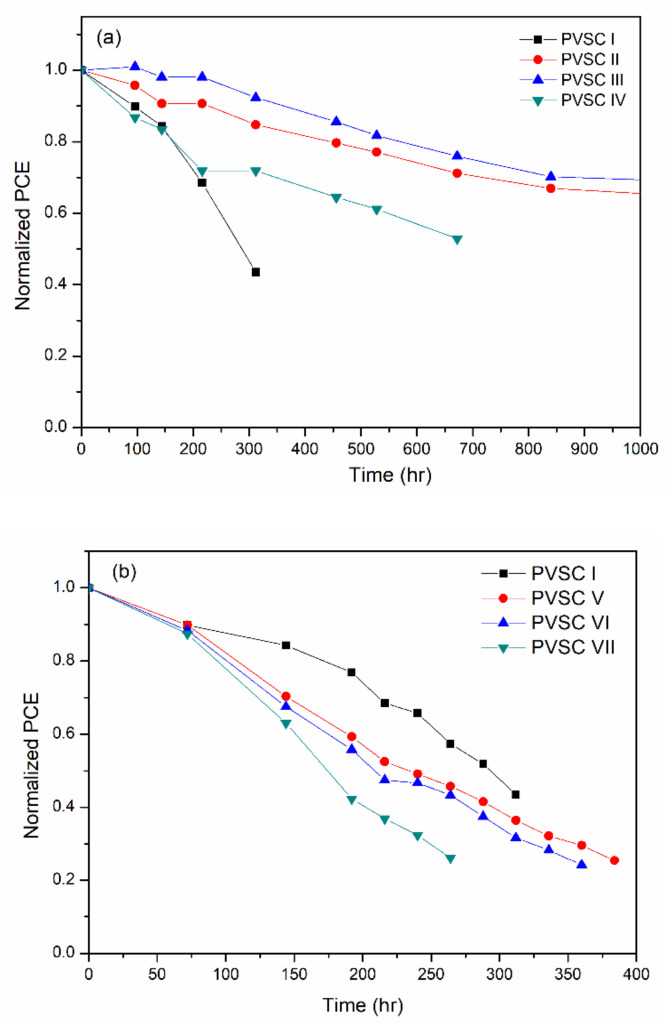
Storage-stability of PVSCs incorporating BCP, BCP/PMMA, and BCP/PVP films ((**a**) BCP (PVSC I), BCP/PMMA (PVSC II-IV); (**b**) BCP/PVP (PVSC V-VII); measured at 30 °C and 35% relative humidity).

**Table 1 polymers-13-00042-t001:** PV properties of PVSCs incorporating BCP, BCP/PMMA, and BCP/PVP films.

PVSC	Interfacial Layer	Composition (*w*/*w*)	*V*_OC_ (V)	*J*_SC_ (mA cm^−2^)	FF	PCE (%)	Best PCE (%)	J_SC_ * (mA cm^−2^)
PVSC-I	BCP	0.0	0.89 ± 0.01	21.18 ± 0.02	0.61 ± 0.01	11.50 ± 0.28	11.78	18.56
PVSC-II	BCP/PMMA	10:1	0.89 ± 0.01	20.69 ± 0.01	0.61 ± 0.01	11.23 ± 0.38	11.61	18.08
PVSC-III	BCP/PMMA	5:1	0.89 ± 0.01	20.92 ± 0.02	0.60 ± 0.01	11.17 ± 0.39	11.56	18.07
PVSC-IV	BCP/PMMA	5:2	0.89 ± 0.01	18.49 ± 0.01	0.63 ± 0.01	10.37 ± 0.29	10.66	17.56
PVSC-V	BCP/PVP	10:1	0.91 ± 0.01	21.33 ± 0.02	0.61 ± 0.01	11.84 ± 0.23	12.07	18.82
PVSC-VI	BCP/PVP	5:1	0.91 ± 0.01	21.71 ± 0.01	0.60 ± 0.02	11.85 ± 0.56	12.41	19.14
PVSC-VII	BCP/PVP	5:2	0.91 ± 0.02	20.45 ± 0.01	0.61 ± 0.01	11.35 ± 0.35	11.70	18.27

* J_SC_ values of the PVSCs calculated from EQE spectra.

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
