# Peer review of "Enhanced Photovoltaic Properties of Perovskite Solar Cells by Employing Bathocuproine/Hydrophobic Polymer Films as Hole-Blocking/Electron-Transporting Interfacial Layers"

_polymers, 2020, doi:10.3390/polym13010042_

Round 1

Reviewer 1 Report

1.In Table 1,the authors should add the standard deviation or error bar of photovoltaic parameters. 2.The degradation mechanisms of cells should be further discussion. As the intrusion of water, oxygen, and Ag into the MAPbI3 may degrade the cells performance. 3.What about the influence of PMMA, PVP on the electron transport of BCP/PCBM. 4.The integrated Jsc from EQE should be added and compared with the Jsc extracted from JV curves. 5.In Fig.12, which parameter(Jsc or Voc) dominated the PCE degradation of cells, further explaining is suggested. 6.It can be seen that, the introducing of PMMA improved the stability but the PCE decreased, while the employing of PVP showed higher PCE but faster degradation. What about the blend of BCP/PMMA/PVP? 7.What about the active area of cells? The baseline PCE is relatively low(

Reviewer 2 Report

The manuscript describes the “Enhanced photovoltaic properties of perovskite solar cells by employing bathocuproine/hydrophobic polymer films as hole-blocking/electron-transporting interfacial layers”. The structural, morphological, and photovoltaic properties of the films were studied and compared with others. After careful evaluation of the paper, I recommend publication subject to a minor revision in the following aspects.

  1. Besides, the manuscript wasn't prepared carefully as shown in the following examples.
  2. Unnecessary upper case, e.g. PVSCs, bathocuproine, etc.
  3. Acronyms in the parentheses, such as BCP, PMMA, BCP/PMMA, PVP, and BCP/PVP should be put right after instead of before the full name to avoid confusion.
  4. Wrong unit/typo, e.g. Degrees (Degree) labels in Fig. 2
  5. The XRD of the BCP are not clear. Why peaks shifting at 2θ= 20⁰, 32 degree.
  6. If possible, XPS or elemental EDX mapping of samples were inserted in the paper.

4 . All figures are not clear. Improve the quality of the figure.   

  1. Provide the crystallite size for all samples separately and the JCPDS cards no. provided in fig. but also provide in text.
  2. Authors have to remove all the typos and grammatical errors in the paper which may interfere with the reading and understanding of the paper.
  3. In the AFM morphology section, the author should mention any relation of nanostructure with PEC application, which is useful for solar cell application.  However, the crystalline of the sample became better, the porous morphology has gradually vanished, and the PEC performance became enhanced. What is the reason for this issue?
  4. Please correct some typos.
  5. In Figure 12 a, b, please explain the decrease in variation of Storage-stability after 50hr for all samples.
